# Evaluation of Soil Moisture and Shear Deformation Based on Compression Wave Velocities in a Shallow Slope Surface Layer

**DOI:** 10.3390/s19153406

**Published:** 2019-08-03

**Authors:** Shangning Tao, Taro Uchimura, Makoto Fukuhara, Junfeng Tang, Yulong Chen, Dong Huang

**Affiliations:** 1Department of Civil and Environmental Engineering, Saitama University, Saitama 338-8570, Japan; 2Faculty of Engineering, Saitama University, Saitama 338-8570, Japan; 3State Key Laboratory of Hydro science and Engineering, Tsinghua University, Beijing 100084, China; 4Institute of Mountain Hazards and Environment, Chinese Academy of Sciences, Chengdu 610041, China

**Keywords:** slope failure, early warning, wave propagation, compression wave velocity, shear deformation, artificial rainfall, multi-layer shear model

## Abstract

Rainfall-induced landslides occur commonly in mountainous areas around the world and cause severe human and infrastructural damage. An early warning system can help people safely escape from a dangerous area and is an economical and effective method to prevent and mitigate rainfall-induced landslides. This paper proposes a method to evaluate soil moisture and shear deformation by compression wave velocities in a shallow slope surface layer. A new type of exciter and new receivers have been developed using a combination of micro electro-mechanical systems (MEMS) accelerometers and the Akaike’s information criterion (AIC) algorithm, which can automatically calculate the elastic wave travel time with accuracy and reliability. Laboratory experiments using a multi-layer shear model were conducted to reproduce the slope failure. The relationships between wave velocities and soil moisture were found to be dependent on the saturation path (rain or drain); in other words, hysteresis was observed. The wave velocity ratio reduced by 0.1–0.2 when the volumetric water content (VWC) increased from 0.1 to 0.27 m^3^/m^3^. When loading the shear stress corresponding to slope angles of 24, 27, 29, or 31 degrees, a drop of 0.2–0.3 in wave velocity ratio was observed at the middle layer, and near 0.5 at the bottom layer. After setting the shear stress to correspond to a slope angle of 33 degrees, the displacement started increasing and finally, slope failure occurred. With increasing displacement, the wave velocities also decreased rapidly. The wave velocity ratio dropped by 0.2 after a displacement of 3 mm. Monitoring long-term elastic wave velocities in a slope surface layer allows one to observe the behavior of the slope, understand its stability, and then apply an early warning system to predict slope failure.

## 1. Introduction

Rainfall-induced landslides commonly occur in mountainous areas and cause severe human and infrastructural damage around the world, in places like Hong Kong [1], USA [2], and Italy [3]. Most of the previous landslides have occurred at shallow depths, generally less than 3 m, and the average thickness of the failed surface layer was 1.2 m [4]. To mitigate damage caused by rainfall-induced landslides, physical countermeasures [5] such as retaining walls [6], ground anchors [7] and dewater systems [8] are common, however, they are not economically feasible for the amount of potentially unstable slope. Therefore, landslide early warning systems are an alternative soft countermeasure that can provide an efficient and economical way to reduce the damage of landslides. A typical landslide early warning system is based on monitoring of soil moisture and pore pressure [2], or on measuring mass movement events by linear displacement transducers [9], inclinometers [10] or extensometers [11], or measuring both the soil moisture and the displacement by soil moisture sensors and tilt sensors [12]. These methods have recently been used because they are simple and easy to install in the slope surface layer. However, they can only sense the local area surrounding the position of the sensor. To cover a wide area of unstable slope, many sensors are required [13].

In this paper, a method of evaluating slope shear deformation and soil moisture by elastic wave velocities is presented. Elastic wave devices include an exciter and several receivers that are laid out within the slope surface layer to cover a relatively deep and wide area, as shown in Figure 1.

Elastic wave propagation in soil as a non-destructive monitoring technique has received considerable attention in recent years. The application of elastic wave propagation in soil has been developed by many researchers, for example, shear waves were measured in laboratory specimens by means of piezoelectric transducers [14], and recently, both shear wave (S-wave) and compression wave (P-wave) velocities were designed to measure the unsaturated soil [15]. It was found that both P-wave and S-wave velocities decreased by nearly half when soil saturation was increased from 20% to 80% in laboratory triaxial experiments [16]. A series of model experiments found that elastic wave velocities continuously decreased in response to moisture content and deformation [17]. To extend the former research, three main points have been improved in this study. Firstly, an exciter has been developed that can automatically generate clear and powerful elastic wave signals to propagate more than 1 m in soil. Secondly, an algorithm has been developed that can process the huge number of wave signals, and automatically detect the travel time and calculate the wave velocities. Thirdly, a full-scale multi-layer shear model was used to simulate the process of slope failure and observe the wave propagation. The detailed behavior of elastic wave propagation in soil over a wide range of soil moisture, shear stress and shear deformation, can be explored. A series of tests were designed to reproduce the slope failure. The corresponding test results and discussion make the mechanism of elastic waves and its practical application clear.

## 2. Testing Apparatus and Devices

The testing apparatus used in this study is a multi-layer shear model. The concept of a multi-layer shear model is shown in Figure 2. The size of the model is based on an investigation of slope failure in Japan, which shows that the average depth of shallow slope failure was 1.2 m [18]. The model can be visualized as a part of the soil cut out from an infinitely long slope surface layer. Assuming that the slope angle is θ, the horizontal direction force can be expressed by tanθ times the weight of the soil. If this 1 m model is divided into a multi-layer model and horizontal force is loaded on every layer, every layer can undergo shear deformation independently.

The multi-layer shear model is shown in Figure 3. It includes 20 layers with a total height of 1 m, where each layer is an independent frame with a height of 0.05 m, length of 0.6 m and width of 0.54 m. Each frame is equipped with wheels and is movable under the horizontal shear force. Shear force is applied on every frame by an air cylinder to simulate the shear force corresponding to the slope angle. Displacement meters are also placed at every layer. Dis1~Dis19 are the displacement meters used to record the shear displacement.

The artificial rainfall system includes an air compressor, pressure regulator, water tank, pipeline and hydraulic spray nozzle. The water tank cover is airtight, and the air pressure is applied above the water surface. A pipeline is connected to the bottom of the tank and the nozzle at the end of the pipeline. Water sprays out under the applied air pressure. A uniform rainfall intensity of 60 mm/h was used in this study. The nozzle is an SSXP series manufactured by H. IKEUCHI and Co., Ltd., Osaka, Japan. Rainwater infiltrates into the top layer and then into sublayer, and finally drains from the bottom of the model. The soil moisture sensor (ECH2O EC-5 (METER Group, Inc., Pullman, WA, USA)) determines the volumetric water content of the soil. Ten soil moisture sensors were set up in the soil, with a vertical interval of 100 mm, to measure the soil moisture distribution with depth.

The newly designed exciter is used to generate pulse elastic waves, as shown in Figure 4a. It has a height of 33 mm, and a diameter of 25 mm. It has a steel ball inside that weighs 16.95 g. It can be pulled up by an electromagnet and then freefalls from a height of 8.3 mm when the electromagnet is released. The receiver is used to sense the pulse elastic waves, as shown in Figure 4b. It has a height of 5 mm, length of 22 mm, and a width of 12 mm. It has a MEMS sensor inside, named ADXL354 (Analog Devices, Norwood, MA, USA), which is a high sensitivity, ultralow noise density, low drift 3-axis accelerometer. A total of 10 exciters and 30 receivers are set in the specified position in the soil. E1~E10 are the exciters used to generate elastic waves; CH01~CH30 are the receivers used to sense the elastic waves.

The design of the multi-layer shear model makes it easy to set up the experiment and easy to understand the detailed behavior of the soil over time until slippage occurs on the slope. Compared to the conventional small direct shear test, the multi-layer shear model is a larger model. Not only is the effect of the sensors inside small, but the behavior up to slope failure can also be analyzed from various viewpoints. It can be easily used to simulate a part of the natural shallow slope surface layer.

## 3. Methods

### 3.1. Elastic Wave Velocities

Figure 5 shows the method to calculate the elastic wave velocity. The wave signal is generated by an exciter installed at the top layer. The wave signal travels through the soil and is detected by receivers. Figure 5 shows that the ch4, the farthest from the exciter, can detect a clear wave signal. The vertical survey line and the vertical axis of the receivers were analyzed. The wave can be considered a compression wave (P-wave) because the survey line is in the compression direction [19]. Several compression wave velocities on the vertical survey line were investigated, and, due to soil moisture, shear stress and shear displacement, are different in these block areas. The wave velocity is expressed by
(1)$$V_{i}=\frac{D_{i}}{T_{i}}=\frac{D_{i}}{t_{i+1}-t_{i}}\;(i=1,2,3,4)$$
where *V_i_* is the wave velocity, *D_i_* is the distance, *t* is the first travel time of the wave signal, *i* is the number of the receiver.

### 3.2. Automatic Travel Time Picking

In this study, 30 waveforms can be measured by 30 channels with a sampling rate of 100 kHz. A total of 2,592,000 waveforms can be collected in a day’s wave monitoring test. Picking the first travel time manually is very time consuming, and it is impossible to adopt this method to analyze the huge amount of data. Automatically picking the travel time by computer is necessary for analyzing the data.

Many researchers have developed techniques and methods for automatically measuring the first travel time of a wave. For example, a computer-based system dedicated full time to automatic detection and location of local seismic events [20] and automatic detection of P-wave and S-wave travel times of small earthquakes [21] has been developed. Reliable automatic picking of the first travel time of acoustic emissions and ultrasound signals in concrete has also presented [22]. It showed that different versions of automatic picking methods appeared due to the different characteristics of the wave signal.

Depending on the wave signal generated by the newly designed exciter in this study, the algorithm for automatic picking of the travel time is based on the Akaike information criterion (AIC) [23,24], which can provide accurate and reliable travel time determination of elastic wave signals.

A waveform with a length of *N* samples can be divided into two parts at a point *k*, shown in Figure 6. The first part includes *k* samples and its normal distribution is *σ*_1_; the second part includes (*N − k*) samples and its normal distribution is *σ*_2_. The AIC value at *k* is defined by the following formula
(2)$$AIC\left(k\right)=k*log({\sigma_{1}}^{2})+\left(N-k\right)*\log({\sigma_{2}}^{2}).$$

The normal distribution *σ* is defined as
(3)$$\sigma^{2}=\frac{1}{n-1}\overset{n}{\underset{i=1}{\sum }}{\left(x_{i}-\mu \right)}^{2}$$
where *n* denotes the length of the signal, $x_{i}$ is sample i of the time series, and μ is the mean value of the whole time series x, defined as
(4)$$\mu =\frac{1}{n}\sum_{i=1}^{n}x_{i}.$$

The AIC value can be calculated by Equation (2) when setting the index *k* from 1 to *N*. The minimum is the first travel time of the elastic wave, shown in Equation (5). In Figure 7, the AIC value is represented by the dashed line, the minimum AIC value denotes the first travel time of the signal.
(5)Travel time=Min{AIC(1),AIC(2),…,AIC(N−1),AIC(N)}

The reliability and accuracy of the determination of the travel time of the elastic wave propagation in soil are important because the travel time is the premise for the interpretation of the corresponding results, and therefore of evaluating the instability of the slope surface layer. In order to examine the reliability of Equation (5), 100 waveforms were randomly extracted from the experiment, and the travel times were calculated by both the AIC algorithm and manually. The result is shown in Figure 8. The arrival times picked by the AIC algorithm and those picked manually were almost the same. This shows that the AIC algorithm can provide a highly accurate and reliable determination of the travel time of elastic wave signals.

## 4. Test Material and Test Procedure

### 4.1. Test Material

The test material used in this study was composed of silica sand No. 4, No. 5, No. 7 and No. 8 mixed at a ratio of 1:1:3:1 to closely resemble the particle size distribution in the natural soil of the slope. It had a dry density of around 1.481 g/cm^3^. Its minimum and maximum dry density were found to be 1.308 g/cm^3^ and 1.707 g/cm^3^. The relative density (Dr) was 50% and the volume water content was 7.4% at the initial state (all tests were conducted according to Japanese Geotechnical Society (JGS) standards; JGS 0161 test method for minimum and maximum densities of sands (JIS A1224)). Figure 9 shows the grain size accumulation curve of the test material.

### 4.2. Calibration of Soil Moisture Sensors

Soil moisture sensors (EC-5) were used to determine volumetric water content (VWC). Soil-Specific calibration is recommended for the best possible accuracy in volumetric water content measurements. Calibration of the EC-5 sensors has been shown to result in an increased accuracy of 1%–2% for all soils with soil-specific calibration [25]. Research has indicated that soil-specific calibration of soil moisture sensors achieves a good performance [26]. For the purpose of attaining a higher accuracy, all of the EC-5 sensors were calibrated with the test material.

### 4.3. Test Cases and Conditions

To simulate the rainfall-induced slope failure, a series of test cases were conducted. These test cases focused on soil moisture, shear stress and shear displacement, which are the main factors that affect slope stability. In these experiments, elastic wave velocities were used to determine the soil moisture, shear stress, and shear displacement, in order to study their behaviors in the process of slope failure. The overall test program was divided into seven series of test cases, and the conditions of every test case are summarized in Table 1.

The specimens were prepared by wet tamping at every 25 mm to construct the model. Test case 1 contained the initial amount of soil moisture, and there was no shear force applied to any layer of the model. After 21 h of rainfall, the soil moisture in the model became stable. Then the rainfall was stopped, and the water was drained out for 46 h. The maximum wave velocities observed in several different survey lines, defined in Figure 5, are named with Vp(initial), which was used as a comparison standard value to express the change ratio of velocities by the following test.

Test case 2 was used to confirm the effect of the loading and unloading shear force on the wave velocities. Test case 3 was used to analyze the effect of soil moisture on the wave velocities during the rainfall and drain events without a loading shear force. Test cases 4, 5, and 6 were used to study the effect of shear stress on the wave velocities during the rainfall and drain events, without shear displacement. Test case 7 was used to find out the relationship between shear displacement and wave velocities under constant shear stress before slope failure. Soil moisture was constant during this test. Figure 10 shows the test procedure over time. The rainfall or drain condition, the shear stress corresponding to the slope angle and the VWC on the top slope is shown.

The distribution of soil moisture with depth is shown in Figure 11a. The data observed by soil moisture sensors helped us understand the changes of the VWC at every layer. Four hours of rainfall could make the VWC at every layer become stable, similar to the results of a 22-h rainfall. High soil moisture content at the bottom was observed in rainfall events. In the drain events, after a long time draining water, the VWC significantly decreased near the top, but a higher VWC was observed near the bottom. These soil moisture distributions can help explain the soil moisture distribution in a natural slope.

Shear stress corresponding to slope angle was different at every layer because the soil weight increased with the depth. The greater the depth, the stronger the shear stress that was applied, shown in Figure 11b.

High soil moisture was observed at the bottom of the model in both rainfall and drain events. High soil moisture decreases the soil strength when a high shear stress is applied. Therefore, the layers near the bottom failed more easily than the other layers.

## 5. Results and Discussions

### 5.1. The Effects of Soil Moisture on Elastic Wave Velocities without Shear Force

The main purpose of test case 3 was to find out the relationship between the wave velocities and soil moisture without shear force applied at any layer. Before test case 3, water in the soil was drained for 288 h. There was subsequently very low soil moisture in the layer near the top of the model. Then test case 3 was conducted with 4 h of rainfall and 19 h of drain water. The amount of soil moisture at different layers after the rainfall and drain events is shown in Figure 11a. The response of elastic wave velocities at different VWC values is summarized in Figure 12.

Figure 12a shows the changes in VWC during the rainfall and drain events over time. During the rainfall event, the VWC near the top surface increased from 0.1 to 0.25 m^3^/m^3^, whereas the VWC near the bottom increased from 0.27 to 0.31 m^3^/m^3^. The response of the wave velocities to the rainfall and drain events over time is shown in Figure 12b. The wave velocity ratio increased at the beginning of the rainfall, shown by the number 1. That is, when the rainwater infiltrated into the area above the wave measure area, the soil moisture content of the upper layer increased and made the vertical compressive stress increase, resulting in an increase of the wave velocity ratio. This is similar to the behavior of Vp during unloading of isotropic stress in the element test [16].

When the rainwater started to infiltrate the wave measurement area, the wave velocity ratio decreased as the soil moisture increased, shown by the number 2. A gradual increase in softening of the soil specimen upon water infiltration may be responsible for the decreasing wave velocities. This decrease continued until the soil moisture of the specimens became stable. On the contrary, the wave velocity ratio increased with decreasing soil moisture during the drain stage, shown by the number 3. Figure 12c shows the wave velocities against the VWC using the same data as Figure 12a,b. A clear relationship between the VWC and the wave velocity ratio can be observed. The wave velocity ratio reduced by 0.1~0.2 when the VWC increased from 0.1 to 0.27 m^3^/m^3^. This is an approximately linear relationship between wave velocities and soil moisture. Compared to the results of the element test [16] and the model test [27], a similar trend of the wave velocities and soil moisture was confirmed. The wave velocities change with soil moisture with a character of near linearity, which is a feature that can be used in the landslide early warning system.

Soil moisture is the key factor of a slope’s stability in the context of rainfall-induced slope failure, which has been concluded by many researchers [1]. When rainwater infiltrates unsaturated soil, the soil moisture will increase. This will lead to a decrease in soil matric suction, resulting in the soil losing its strength. Elastic wave propagation in soil is a geometric spreading; the wave velocities depend on the soil strength. The effects of soil moisture on the wave velocities can be explained by matric suction. The changes in soil moisture content can be expressed by the changes in matric suction; high soil moisture content means lower matric suction, and the weaker force between soil partials results in lower wave velocities. There is hysteresis in the path of wave velocities and the VWC between the rainfall and drain events (Figure 12c). This may be related to the hysteresis observed in the relationship between soil moisture and matric suction.

### 5.2. The Effects of Shear Stress on Elastic Wave Velocities

If the loads on the slope change, the shear stresses within the soil will change. Shear stresses also change due to earth pressure changes with different levels of soil moisture or soil deformation, resulting in an unstable slope or slope failure. To understand the effect of loading and unloading shear force on elastic wave velocities, the survey line between ch12 and ch13 was analyzed (Figure 3b). The test cases of 2, 4, 5, and 6 in Table 1 were included. The results are shown in Figure 13. This shows that when loading the shear stress so it corresponds to a 24-degrees slope angle, the Vp/Vp(initial) ratio dropped from 0.85 to 0.65, where the VWC = 0.19. On the contrary, when unloading the shear stress from 24-degrees to 0-degree, the wave velocities ratio increased from 0.6 to 0.8, where the VWC = 0.28. Furthermore, when loading the shear stress so it corresponds to a 27-degrees slope angle or higher, the Vp/Vp(initial) ratio reduced to 0.62, or below 0.6. This shows that the stronger the shear stress, the lower the wave velocities ratio. It can be considered that this is because the resistance force against the shear in the specimens changes the direction of wave propagation, resulting in a reduction of the vertical wave velocity. We attempted to find related research, but most of it focused on how the vertical compressive stress affects the wave velocity in the element test. No previous research directly focused on the effect of shear stress on wave velocity.

The changes in wave velocity ratio with the VWC in loading shear stress is parallel with the unloading shear stress. That is, the relationship between wave velocities and shear stress is independent. This shows the advantage of wave velocities in reflecting VWC and shear stress, meaning they are suitable for predicting the stability of the slope.

Figure 11 shows that not only the distribution of soil moisture but also the shear stress vary with depth in the model. The effects of shear stress on elastic wave velocities in several areas were investigated. These areas were at different depths in the model, located in the vertical survey line under the excitor (E1). The results are shown in Figure 14 and Figure 15. Figure 14a shows the response of wave velocities at different shear stress levels during rainfall events. Figure 15a shows the response of wave velocities at different shear stress levels during drain events. Figure 14b and Figure 15b show the wave velocity ratio against the shear stress at every layer with the same soil moisture content. The relationship between the wave velocity ratio and the shear stress is near linear. This shows that the closer to the bottom, the lower the wave velocities. The wave velocity ratio reduced by 0.2~0.3 at the middle layer and about 0.5 near the bottom where the shear stress is highest.

In the natural slope, shear stresses increase by many factors like water pressure in cracks at the top of the slope, increase in soil weight due to increased soil moisture content, and earthquake shaking. Wave velocities are very sensitive to shear stress. This feature is helpful for monitoring the stability of the slope. Monitoring the wave velocities in a natural slope may detect the changes in shear stress and easily enable one to find the stability of the slope at an early stage.

### 5.3. Elastic Wave Velocities and Shear Displacement

Test case 7 was used to investigate the effect of shear displacement on elastic wave velocities before slope failure. Soil moisture content did not change during this test. High shear stress and high soil moisture content were observed near the bottom (Figure 11), and maximum displacement was observed near the bottom after a shear force corresponding to a slope angle of 33-degrees was applied. Therefore, the displacement of the sensor Dis18 and the velocities at the vertical survey line between receivers ch13 and ch4, which are near the bottom of the model, were analyzed. Figure 16 shows the wave velocities and displacement plotted over time. When the shear force corresponding to a slope angle of 32-degrees was set, a very small displacement appeared but stopped moving after 2 h. When the shear force corresponding to a slope angle of 33-degrees was set, the slope started moving with an average speed of 3 mm/h, then accelerated and finally failed. The model showed that with increased displacement, the wave velocities decreased rapidly. The displacement–time relationship before failure of the soil, defined as the creep of the soil [28], could be observed in this test. Therefore, wave velocities can be used to detect the creep of the soil.

The same wave velocities and displacement data in Figure 16 were plotted in Figure 17. This figure shows that there is a linear relationship between wave velocities and displacement. The wave velocities changed from 105 m/s to 75 m/s when the displacement developed from 7 mm to 10 mm. That is, the wave velocity ratio dropped by 0.2 after a displacement of 3 mm. This shows that elastic wave velocities are also sensitive to shear displacement. Therefore, elastic wave velocities can be used to detect the slope when it starts to move and send out an alarm to warn of the dangerous situation of the slope before it fails.

Measuring the shear displacement by linear displacement transducers [9], inclinometers [10], extensometers [11] or tilt sensors [12] are the current methods of predicting slope failure. As the elastic wave is sensitive to soil moisture, shear stress and shear displacement, it may be a new method for predicting slope failure.

## 6. Conclusions

A new type of exciter and receiver have been developed by assembling a MEMS accelerometer and using the AIC algorithm, which can automatically calculate the elastic wave travel time with accuracy and reliability. Laboratory experiments using a multi-layer shear model simulating shallow slope failure were conducted to observe the changes in elastic wave propagation in a slope surface layer. The results can be concluded as follows:(1)Effects of soil moisture on elastic wave velocities. The wave velocities decreased with increasing soil moisture in the rain event and increased during the drain stage. The wave velocity ratio reduced by 0.1–0.2 when the volume of water content increased from 0.1 to 0.27 m^3^/m^3^.(2)Effects of shear stress on elastic wave velocities. The stronger the shear force applied, the lower the velocities observed. When loading a shear stress corresponding to slope angles of 24, 27, 29, and 31 degrees, a drop in wave velocity of 0.2–0.3 was observed at the middle layer, and near 0.5 at the bottom layer.(3)Effects of shear displacement on elastic wave velocities. Increasing the displacement caused the wave velocities to also increase. The wave velocity ratio dropped by 0.2 after 3 mm of displacement.

Monitoring the wave propagation in a slope surface layer can indicate the status of soil moisture content and shear deformation. Slope instabilities may be predicted based on the historical record. Monitoring the changes in elastic waves in the slope surface layer is valuable and can be applied to an early warning system.

## Figures and Tables

**Figure 1 sensors-19-03406-f001:**
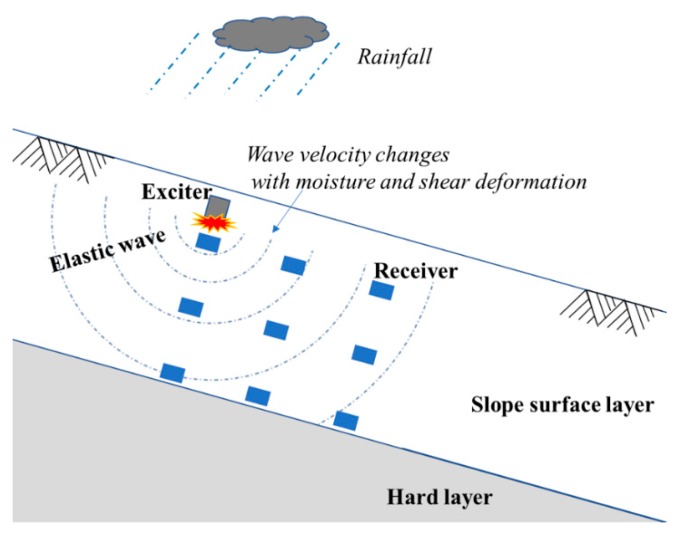
The typical layout of the sensors to determine the soil moisture and shear deformation by elastic wave propagation in a slope surface layer.

**Figure 2 sensors-19-03406-f002:**
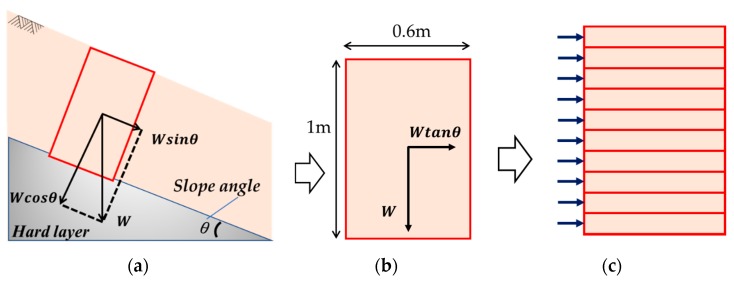
The concept of the multi-layer shear model used in the lab experiment. (**a**) A part of the soil cut out from an infinitely long slope surface layer. The horizontal direction force can be expressed by tanθ times the weight of the soil. (**b**) The force ratio is the same as it turned to the horizontal direction. (**c**) The 1 m model is divided into the multi-layer model, the horizontal force is loaded on every layer, and every layer can undergo shear deformation independently.

**Figure 3 sensors-19-03406-f003:**
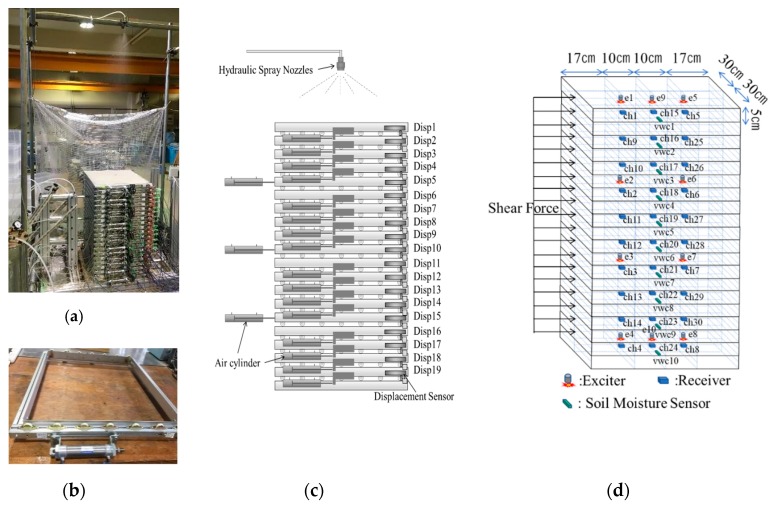
Multi-layer shear model and sensors layout. (**a**) A photo overview of the multi-layer shear model; (**b**) an independent frame of the multi-layer shear model; (**c**) the layout of air cylinders and displacement meters, Dis1~Dis19 are the displacement meters; (**d**) the layout of sensors in the soil. E1~E10 are the exciters; CH01~CH30 are the receivers; VWC1~VWC10 are the soil moisture sensors.

**Figure 4 sensors-19-03406-f004:**
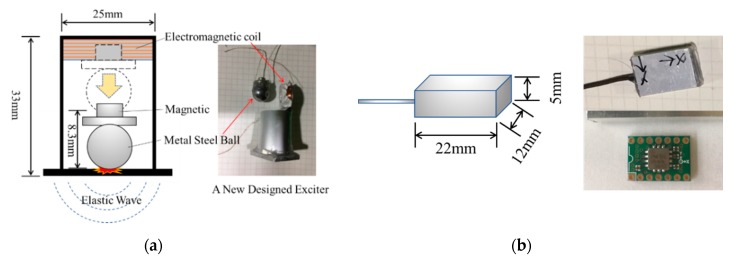
Newly designed exciter and receiver: (**a**) The newly designed exciter is used to generate pulse elastic waves. It has a height of 33 mm and a width of 25 mm. The mass of the steel ball is 16.95 g. It can be pulled up by the electromagnet and freefalls from a height of 8.3 mm when the electromagnet is released. (**b**) The receiver has a height of 5 mm, length of 22 mm, and width of 12 mm. A MEMS three axle accelerometer is inside.

**Figure 5 sensors-19-03406-f005:**
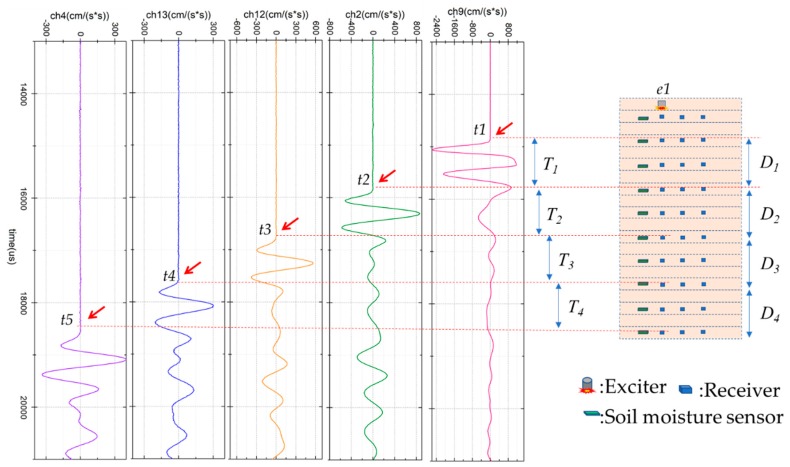
Waveforms and the method to calculate the elastic wave velocity. The wave signal is generated by exciter e1 and detected by receivers in the vertical survey line. The wave velocity is defined by the travel distance divided by the travel time between two receivers. t1~t5 are the first travel times of the wave signals.

**Figure 6 sensors-19-03406-f006:**
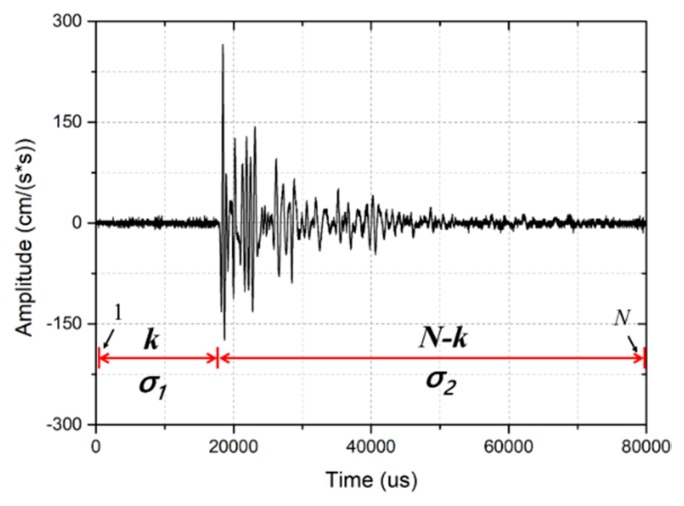
A time series of a wave signal is divided into two parts, the first part includes k samples and its normal distribution is *σ*_1_; the second part includes (N– k) samples and its normal distribution is σ_2_.

**Figure 7 sensors-19-03406-f007:**
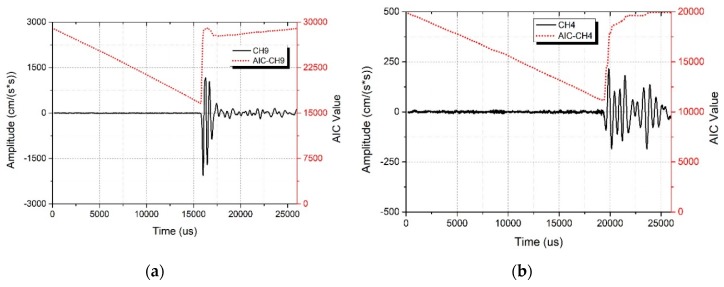
The waveform and AIC value. The AIC value is represented by the dashed line, the minimum AIC value denotes the first travel time. (**a**) Strong signal near the exciter, (**b**) weak signal far away from the exciter. Note the different scale on the y-axes in (a) and (b).

**Figure 8 sensors-19-03406-f008:**
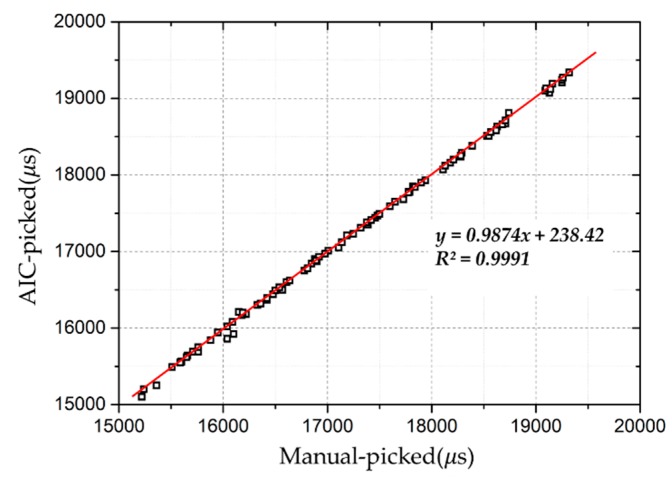
Travel times picked by the AIC have almost the same results as those picked manually.

**Figure 9 sensors-19-03406-f009:**
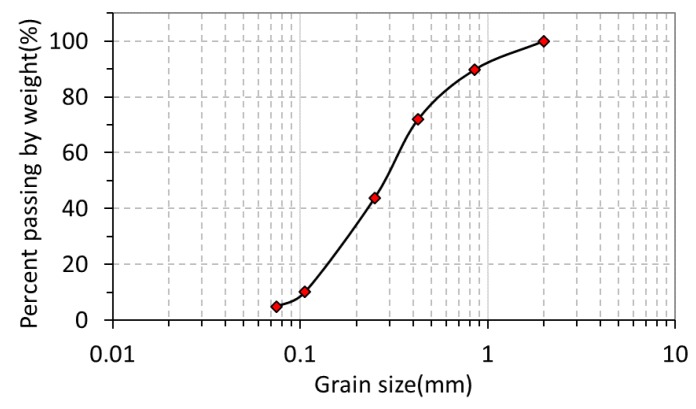
Grain size accumulation curve of the test material.

**Figure 10 sensors-19-03406-f010:**
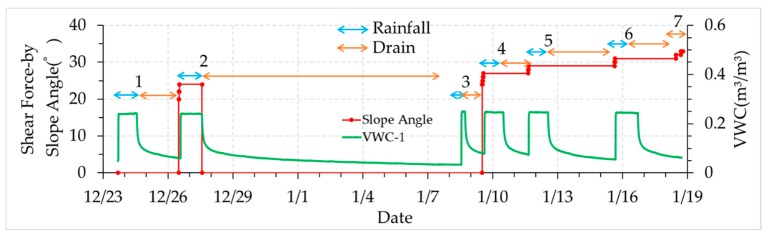
Test cases in the test procedure with a time serial.

**Figure 11 sensors-19-03406-f011:**
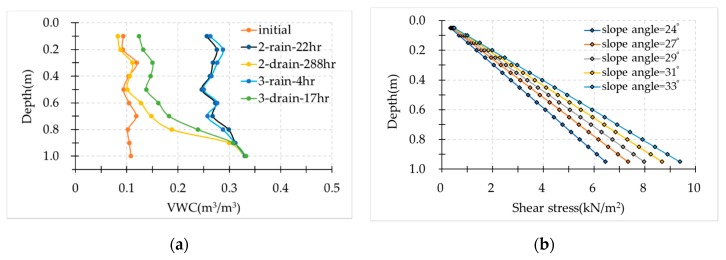
Test conditions. (**a**) Soil moisture distribution with depth after the rainfall and drain events; (**b**) shear stress with a depth corresponding to the slope angle.

**Figure 12 sensors-19-03406-f012:**
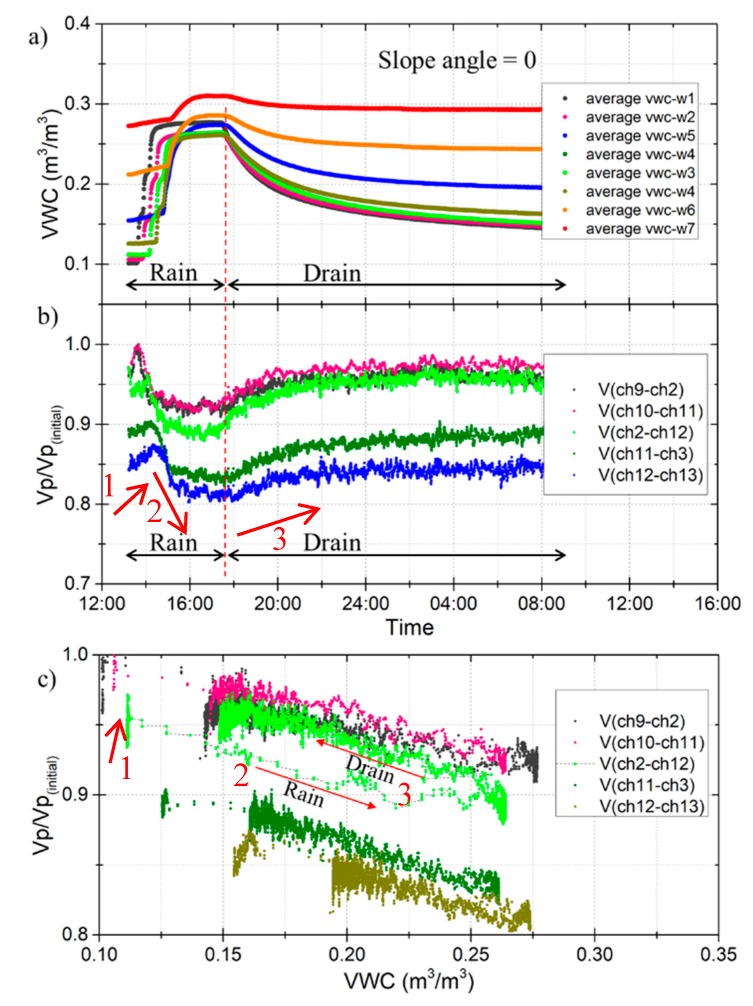
Response of elastic wave velocities at different VWC values during the rain and drain events (slope angle = 0); (**a**) average VWC versus time; (**b**) compression wave velocities (Vp/Vp (initial)) versus time; (**c**) compression wave velocities (Vp/Vp (initial)) versus the VWC.

**Figure 13 sensors-19-03406-f013:**
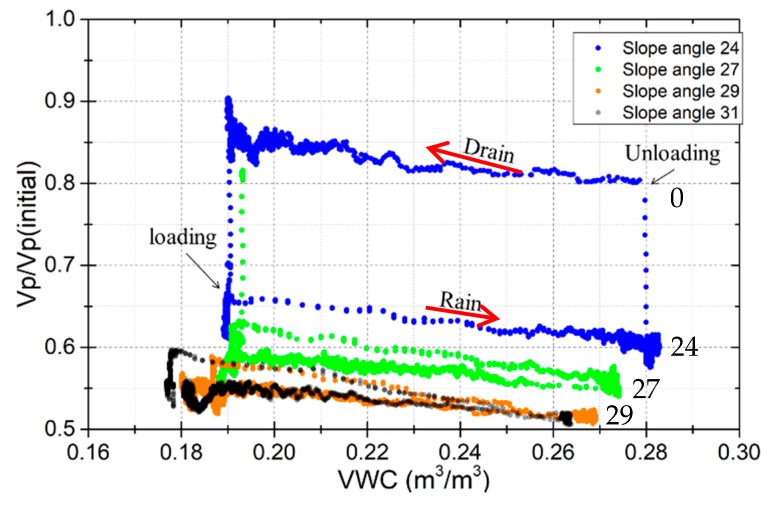
Wave velocities ratio changes with slope angle at the survey line between receivers ch12 and ch13.

**Figure 14 sensors-19-03406-f014:**
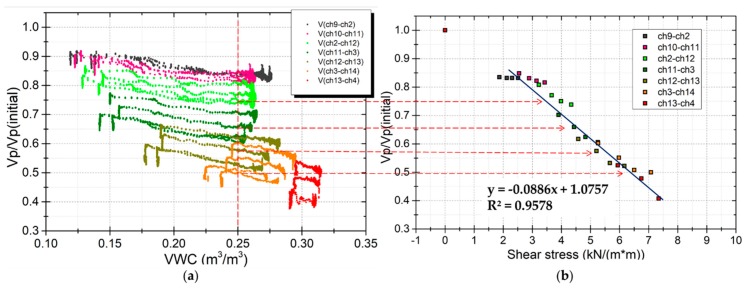
Response of wave velocities at different levels of shear stress during rainfall events. (**a**) Wave velocities ratio against shear stress during the rainfall event. (**b**) Wave velocities ratio response with shear stress at the same 0.25 m^3^/m^3^ of VWC.

**Figure 15 sensors-19-03406-f015:**
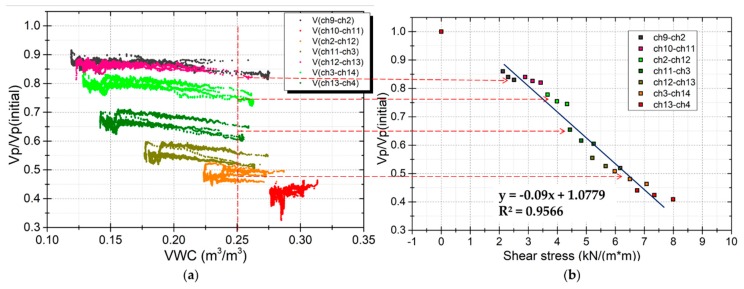
Response of wave velocities at different levels of shear stress during drain events. (**a**) Wave velocity ratio against shear stress during the drain event. (**b**) Wave velocity ratio response with shear stress at the same 0.25 m^3^/m^3^ of VWC.

**Figure 16 sensors-19-03406-f016:**
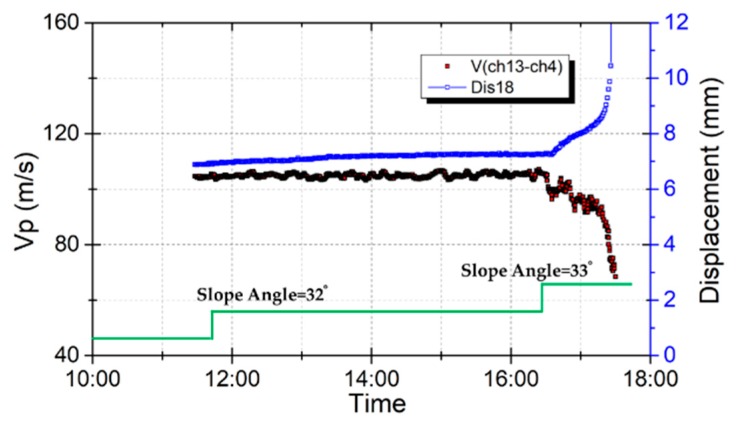
Effect of wave velocities on displacement during an applied shear force.

**Figure 17 sensors-19-03406-f017:**
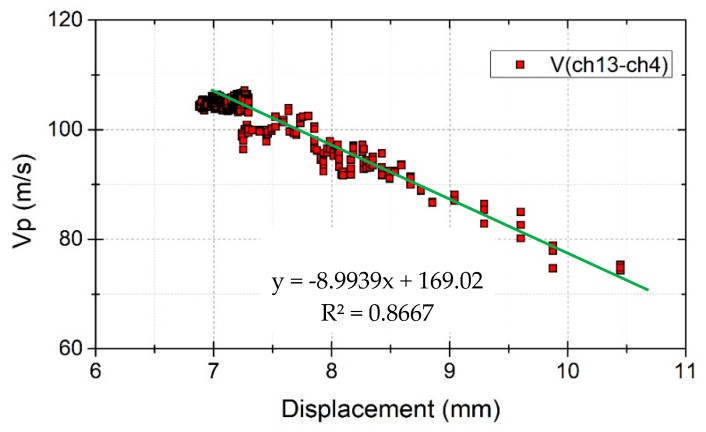
Wave velocity changes with displacement.

**Table 1 sensors-19-03406-t001:** Test cases and conditions.

Test Case	Soil Moisture Control	Shear Force (Slope Angle)	Shear Displacement Observed (mm)
1-1	Rainfall (21 h)	0°	0
1-2	Drain (46 h)	0°	0
2-1	Rainfall (24 h)	24°	0
2-2	Drain (288 h)	0°	0
3-1	Rainfall (4 h)	0°	0
3-2	Drain (19 h)	0°	0
4-1	Rainfall (24 h)	27°	0
4-2	Drain (28 h)	27°	0
5-1	Rainfall (22 h)	29°	0
5-2	Drain (70 h)	29°	0
6-1	Rainfall (23 h)	31°	0
6-2	Drain (41 h)	31°	0
7	Constant	32°, 33°	50

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
