# Peer review of "Evaluation of Soil Moisture and Shear Deformation Based on Compression Wave Velocities in a Shallow Slope Surface Layer"

_sensors, 2019, doi:10.3390/s19153406_

Round 1

Reviewer 1 Report

In the present study the authors present a method to evaluate soil moisture and shear deformation by compress wave velocities in shallow slope surface layer. A new type of exciter and receivers have been developed by assembling Micro Electro Mechanical Systems (MEMS) accelerometers and Akaike's Information Criterion (AIC) algorithm. The approach has been applied to a series of test cases.

In reviewer's opinion the topic of the paper is interesting, but the following issues must be clarified before considering the paper for a possible publication in the journal.

a)  Originality and novelty of the method proposed. This issue must be clarified and well highlighted in the text.

b)  Advantages and limitations of the approach proposed respect to other ones existing in literature, in particular in the applications to real cases. Moreover, the authors could offer a broader view of their method in the context of the available literature on the topic.

c)   Paragraph 7 “Results and Discussion” need to be expanded and discussed in more detailed with a better presentation and discussion of the results obtained.

Other corrections

In the section Introduction (page 2 line 44-45), after the statement “To mitigate damage caused by rainfall-induced landslide, physical countermeasures such as retaining walls, ground anchors and dewater systems are common”for completness the following recent references could be added:

Popescu M.E., Sasahara K. (2009) Engineering Measures for Landslide Disaster Mitigation. In: Sassa K., Canuti P. (eds) Landslides – Disaster Risk Reduction. Springer, Berlin, Heidelberg

Conte, E., Troncone, A., Vena, M. (2017). A method for the design of embedded cantilever retaining walls under static and seismic loading. Géotechnique, Vol. 67, 12, 1081-1089. DOI: 10.1680/jgeot.16.P.201

Conte, E., Troncone, A. (2018).  A performance-based method for the design of drainage trenches used to stabilize slopes. Engineering Geology, Vol. 239, 158-166. DOI: 10.1016/j.enggeo.2018.03.017

Author Response

Response to Reviewer 1 Comments

Point 1: Originality and novelty of the method proposed. This issue must be clarified and well highlighted in the text.

Response 1: Originality and novelty of the method is shown in the following.

p. 2, lines 67-74:” To extend the former research, three main points have been improved in this study. Firstly, an exciter has been developed that can automatically generate clear and powerful elastic wave signals to propagate more than 1 m in soil. Secondly, an algorithm has been developed that can process the huge number of wave signals, and automatically detect the travel time and calculate the wave velocities. Thirdly, a full-scale multi-layer shear model was used to simulate the process of slope failure and observe the wave propagation. The detailed behavior of elastic wave propagation in soil over a wide range of soil moisture, shear stress and shear deformation, can be explored.”

 p. 4, lines 114-118: “The design of the multi-layer shear model makes it easy to set up the experiment and easy to understand the detailed behavior of the soil over time, until slippage occurs on the slope. Compared to the conventional small direct shear test, the multi-layer shear model is a larger model. Not only is the effect of the sensors inside small, but the behavior up to slope failure can also be analyzed from various viewpoints. It can be easily used to simulate a part of the natural shallow slope surface layer.

Point 2: Advantages and limitations of the approach proposed respect to other ones existing in literature, in particular in the applications to real cases. Moreover, the authors could offer a broader view of their method in the context of the available literature on the topic.

Response 2: The view of the countermeasures about rainfall-induced landslides is shown as following.

p. 2, lines 44-57: “To mitigate damage caused by rainfall-induced landslides, physical countermeasures [5] such as retaining walls [6], ground anchors [7] and dewater systems [8] are common, however, they are not economically feasible for the amount of potentially unstable slope. Therefore, landslide early warning systems are an alternative soft countermeasure that can provide an efficient and economical way to reduce the damage of landslides. A typical landslide early warning system is based on monitoring of soil moisture and pore pressure [2], or on measuring mass movement events by linear displacement transducers [9], inclinometers [10] or extensometers [11], or measuring both the soil moisture and the displacement by soil moisture sensors and tilt sensors [12]. These methods have recently been used because they are simple and easy to install in the slope surface layer. However, they can only sense the local area surrounding the position of the sensor. To cover a wide area of unstable slope, many sensors are required [13].

In this paper, a method of evaluating slope shear deformation and soil moisture by elastic wave velocities is presented. Elastic wave devices include an exciter and several receivers that are laid out within the slope surface layer to cover a relatively deep and wide area, as shown in Figure 1.”

Point 3: Paragraph 7 “Results and Discussion” need to be expanded and discussed in more detailed with a better presentation and discussion of the results obtained.

 Response 3: we expand the discussion as the following.

MODIFIED p. 10 lines 260-281.

“The wave velocity ratio increased at the beginning of the rainfall, shown by the number 1. That is, when the rainwater infiltrated into the area above the wave measure area, the soil moisture content of the upper layer increased and made the vertical compressive stress increase, resulting in an increase of the wave velocity ratio. This is similar to the behavior of Vp during unloading of isotropic stress in the element test [16].

 When the rainwater started to infiltrate the wave measurement area, the wave velocity ratio decreased as the soil moisture increased, shown by the number 2. A gradual increase in softening of the soil specimen upon water infiltration may be responsible for the decreasing wave velocities. This decrease continued until the soil moisture of the specimens became stable. On the contrary, the wave velocity ratio increased with decreasing soil moisture during the drain stage, shown by the number 3. Figure 12(c) shows the wave velocities against the VWC using the same data as Figure 12(a),(b). A clear relationship between the VWC and the wave velocity ratio can be observed. The wave velocity ratio reduced by 0.1~0.2 when the VWC increased from 0.1 to 0.27 m3/m3. This is an approximately linear relationship between wave velocities and soil moisture. Compared to the results of the element test [16] and the model test [27], a similar trend of the wave velocities and soil moisture was confirmed. The wave velocities change with soil moisture with a character of near linearity, which is a feature that can be used in the landslide early warning system.”

MODIFIED p. 11 lines 302-307.

“This shows that the stronger the shear stress, the lower the wave velocities ratio. It can be considered that this is because the resistance force against the shear in the specimens changes the direction of wave propagation, resulting in a reduction of the vertical wave velocity. We attempted to find related research, but most of it focused on how the vertical compressive stress affects the wave velocity in the element test. No previous research directly focused on the effect of shear stress on wave velocity.”

MODIFIED p. 13 lines 346-350.

“When the shear force corresponding to a slope angle of 33-degrees was set, the slope started moving with an average speed of 3 mm/h, then accelerated and finally failed. The model showed that with increased displacement, the wave velocities decreased rapidly. The displacement–time relationship before failure of the soil, defined as the creep of the soil [28], could be observed in this test. Therefore, wave velocities can be used to detect the creep of the soil.”

MODIFIED p. 14 lines 363-366.

“ Measuring the shear displacement by linear displacement transducers [9], inclinometers [10], extensometers [11] or tilt sensors [12] are the current methods of predicting slope failure. As the elastic wave is sensitive to soil moisture, shear stress and shear displacement, it may be a new method for predicting slope failure.”

Point 4: In the section Introduction (page 2 line 44-45), after the statement “To mitigate damage caused by rainfall-induced landslide, physical countermeasures such as retaining walls, ground anchors and dewater systems are common” for completness the following recent references could be added:

Popescu M.E., Sasahara K. (2009) Engineering Measures for Landslide Disaster Mitigation. In: Sassa K., Canuti P. (eds) Landslides – Disaster Risk Reduction. Springer, Berlin, Heidelberg

Conte, E., Troncone, A., Vena, M. (2017). A method for the design of embedded cantilever retaining walls under static and seismic loading. Géotechnique, Vol. 67, 12, 1081-1089. DOI: 10.1680/jgeot.16.P.201

Conte, E., Troncone, A. (2018).  A performance-based method for the design of drainage trenches used to stabilize slopes. Engineering Geology, Vol. 239, 158-166. DOI: 10.1016/j.enggeo.2018.03.017

Response 4: We appreciate the comments, add these references for supporting the statement.

MODIFIED p. 2, lines 44 and 46.

“To mitigate damage caused by rainfall-induced landslides, physical countermeasures [5] such as retaining walls [6], ground anchors [7] and dewater systems [8] are common,”

ADDED p. 15, lines 401-406.

5.         Popescu, M.E.; Sasahara, K. Engineering Measures for Landslide Disaster Mitigation. Landslides – Disaster Risk Reduct. 2008, 609–631.

6.         Conte, E.; Troncone, A.; Vena, M. A method for the design of embedded cantilever retaining walls under static and seismic loading. Géotechnique 2017, 1–9.

8.         Conte, E.; Troncone, A. A performance-based method for the design of drainage trenches used to stabilize slopes. Eng. Geol. 2018, 239, 158–166.

Reviewer 2 Report

Review of: Development of Technical Method to Evaluate Soil Moisture and Shear Deformation by Compress Wave Velocities in Shallow Slope Surface Layer, by Shangning Tao et al.

This manuscript details methods of slope modeling and monitoring which may have applications toward long term monitoring of slope stability and potentially the development of a slope failure early warning system.

General Comments

The biggest issue with this manuscript is the language.  In current form, parts of the manuscript are nearly unreadable.  While I can get the gist of what is being said, many of the details are unclear and thus I cannot give the manuscript a proper review.  This is unfortunate because there appear to be good data in the manuscript and this looks like a useful approach.  The figures seem informative (although, I would like a photograph along with Figure 3).

Therefore, at this time, I cannot recommend publication of this manuscript.  However, I do think the article has potential if the language issues are resolved.

I make a few additional comments below, but as I said, I could not give a thorough review.

Specific Comments

Lines 1-3.  The title.  I think "Deformation by Compress Wave Velocities" is not what the authors intend to say.  Is it " Deformation by Compression Waves" instead?

Abstract:  Parts of the abstract are not readable due to plural/singular mistakes or mixing of tenses. 

Lines 27 and 28:  "... Volumetric Water Content (VWC) increased from 0.1~0.27 m3/m3."  instead of m3/m3, consider vol/vol, v/v, or multiply by 100 and use %.  I know m3/m3 is used (as is cc/cc), but technically VWC is a dimensionless value and the reader knows it is the volume number and not the mass. Whatever the authors do (if anything), it should be applied to the entire manuscript.

Lines 41 and 42.  The authors need to pick a tense and use proper plurals.  For example, "Rainfall-induced landslides commonly occur in mountainous areas and cause severe human and infrastructure damage in many countries [1–3]."   This is a problem throughout the manuscript.

Lines 80 and 81.  " imaged"  I think should be "imagined".  The sentence should be rewritten though, "Figure 2a is a cross-section diagram of a slope with angle ?."  Or similar. 

Line 82 and Figure 2 caption:  is 1: tan? meant to be 1/ tan? ?  As in 1/ tan? = cos?/sin? = cot? ? 

Line 85.  Figure 3a, not 4a.

Line 89. "Dis1~dis38" - however, the figure only indicated Dis1 to Dis19.

Line 91 to 92. "A hydraulic spray nozzle is a part of the artificial rainfall system, can provide a uniform rainfall intensity of 60mm/h, controlled by constant air pressure."  How it is controlled by "air pressure" if it is spraying water?

Figure 3.  A photo would better communicate this figure.

Line 93 Should maybe be "ECH2O EC-5 (METER Group, Inc. USA)"

Line 144.  kHz is already in units of "per second" - adding a second "per second" implies the rate is accelerating.

Line 159.  Should be "(N-k)" I believe – so it is not confused with "n" in later formulae.

Figure 7.  Caption should say something like "Note different axes in a) and b)" at the end.  It's a big difference in amplitude so the authors should make that clear.

Line 179.  The R2 is not as important as the slope – which should be near 1 (as it is) and probably tied to 0.

Lines 184 to 186.  This is repeated through the text - and doesn't need to be.  I think the authors only need to say it once.

Line 188.  Please cite the JGS 0161 Test method for minimum and maximum densities of sands (JIS A1224).

214 to 221.  I think these can be combined and reworded into a single paragraph.

Figure 14 caption.  Are the wave velocities causing this?  Or, are they simply changing because of it?  The caption makes it sound like the wave velocities are the cause.

When using Vp/Vp(initial), I would avoid using "%" because it is a ratio and the reader will not catch the reported %'s are based on the Vp(initial).  For example, a drop from 0.85 to 0.65 is a drop of ~24% (0.65 is 74% of 0.85).  And increase of 0.6 to 0.8 is an increase of 33% (0.8 is 133% of 0.6).  If the authors want to use %, then the ratio should be reported as a percent (Vp/Vp(initial) x 100) – then it will be obvious to the reader that reported percent values are based on Vp(initial).  Or better yet, just report the ratio change (0.20)

322 Figure 16??

Line 333.  Is it Figure 17 or 15?

More general comments

Figure 3 and the manuscript text create confusion.   In 3a, it appears that each layer is separated in a tray (frame) with wheels (stated in the text).  I assume that the soil column is inside these stacked frames (like a typical shear stack model) – but this needs to be clearer to the reader because the way 3b is drawn makes it confusing.  This is where a photo might help – and better clarity in the manuscript text.  For any reader not familiar with a ring or shear stack, a mis-impression of the apparatus here can make the entire manuscript confusing.

The authors should also insure that everything mentioned in the abstract is also mentioned in the paper.

Author Response

Response to Reviewer 2 Comments

Point 1: Lines 1-3.  The title.  I think "Deformation by Compress Wave Velocities" is not what the authors intend to say.  Is it " Deformation by Compression Waves" instead?

Response 1: We apologize for such ambiguous expression of the title. We shorten the title as “Evaluation of Soil Moisture and Shear Deformation Based on Compress Wave Velocities in a Shallow Slope Surface Layer”.

Point 2: Lines 27 and 28:  "... Volumetric Water Content (VWC) increased from 0.1~0.27 m3/m3."  instead of m3/m3, consider vol/vol, v/v, or multiply by 100 and use %.  I know m3/m3 is used (as is cc/cc), but technically VWC is a dimensionless value and the reader knows it is the volume number and not the mass. Whatever the authors do (if anything), it should be applied to the entire manuscript.

Response 2: We express the Volumetric Water Content (VWC) by m3/m3 in decimal but not percent because some papers also express the VWC in decimal.

Topp, G.C., J.L. Davis, A.P. Annan (1980): Electromagnetic determination of soil water content. Water Resources Research, 16:574-582

Dorigo, W. A., et al. (2017): ESA CCI Soil Moisture for improved Earth system understanding: State-of-the art and future directions. Remote Sensing of Environment, 203:185- 215.

Point 3: Lines 41 and 42.  The authors need to pick a tense and use proper plurals.  For example,"Rainfall-induced landslides commonly occur in mountainous areas and cause severe human and infrastructure damage in many countries [1–3]."   This is a problem throughout the manuscript.

Response 3: We apologize for such omission.

MODIFIED   p.2 lines 41-42.

p.2 lines 44-47.

p.5 lines 150-156.

Point 4: Lines 80 and 81.  " imaged"  I think should be "imagined".  The sentence should be rewritten though, "Figure 2a is a cross-section diagram of a slope with angle ?."  Or similar.  

Response 4:

CORRECTED p.3 lines 80-81.

“The model can be visualized as a part of the soil cut out from an infinitely long slope surface layer.”

Point 5: Line 82 and Figure 2 caption:  is 1: tan? meant to be 1/ tan? ?  As in 1/ tan? = cos?/sin? = cot?

Response 5:

MODIFIED p.3 lines 82-83.

“Assuming that the slope angle is , the horizontal direction force can be expressed by  times the weight of the soil.”

Point 6: Line 85.  Figure 3a, not 4a.

Response 6:

MODIFIED.

Point 7: Line 89. "Dis1~dis38" - however, the figure only indicated Dis1 to Dis19.

Response 7:

MODIFIED.

Point 8: Line 91 to 92. "A hydraulic spray nozzle is a part of the artificial rainfall system, can provide a uniform rainfall intensity of 60mm/h, controlled by constant air pressure."  How it is controlled by "air pressure" if it is spraying water?

Response 8:

MODIFIED p.3 lines91-95.

“The artificial rainfall system includes an air compressor, pressure regulator, water tank, pipeline and hydraulic spray nozzle. The water tank cover is airtight, and the air pressure is applied above the water surface. A pipeline is connected to the bottom of the tank and the nozzle at the end of the pipeline. Water sprays out under the applied air pressure. A uniform rainfall intensity of 60 mm/h was used in this study. The nozzle is a SSXP series manufactured by H. IKEUCHI & Co., Ltd.”

Point 9: Figure 3.  A photo would better communicate this figure.

Response 9: We add two photos of the apparatus.

MODIFIED p.4 lines 118-123.

 (a) A photo overview of the multi-layer shear model; (b) an independent frame of the multi-layer shear model;

Point 10: Line 93 Should maybe be "ECH2O EC-5 (METER Group, Inc. USA)"

Response 10:

MODIFIED.

Point 11: Line 144.  kHz is already in units of "per second" - adding a second "per second" implies the rate is accelerating.

Response 11:

MODIFIED.

Point 12: Line 159.  Should be "(N-k)" I believe – so it is not confused with "n" in later formulae.

Response 12:

MODIFIED.

Point 13: Figure 7.  Caption should say something like "Note different axes in a) and b)" at the end.  It's a big difference in amplitude so the authors should make that clear.

Response 13:

MODIFIED.

Point 14: Line 179.  The R2 is not as important as the slope – which should be near 1 (as it is) and probably tied to 0.

Response 14:

MODIFIED.

Point 15: Lines 184 to 186.  This is repeated through the text - and doesn't need to be.  I think the authors only need to say it once.

Response 15:

MODIFIED.

Point 16: Line 188.  Please cite the JGS 0161 Test method for minimum and maximum densities of sands (JIS A1224).

Response 16: We appreciate the comments.

MODIFIED.

Point 17: 214 to 221.  I think these can be combined and reworded into a single paragraph.

Response 17:

MODIFIED.

Point 18: Figure 14 caption.  Are the wave velocities causing this?  Or, are they simply changing because of it?  The caption makes it sound like the wave velocities are the cause.

Response 18:

MODIFIED p.12 lines 314-316.

“Figure 14. Response of wave velocities at different levels of shear stress during rainfall events. (a) Wave velocities ratio against shear stress during the rainfall event. (b) Wave velocities ratio response with shear stress at the same 0.25 m3/m3 of VWC.”

Point 19: When using Vp/Vp(initial), I would avoid using "%" because it is a ratio and the reader will not catch the reported %'s are based on the Vp(initial).  For example, a drop from 0.85 to 0.65 is a drop of ~24% (0.65 is 74% of 0.85).  And increase of 0.6 to 0.8 is an increase of 33% (0.8 is 133% of 0.6).  If the authors want to use %, then the ratio should be reported as a percent (Vp/Vp(initial) x 100) – then it will be obvious to the reader that reported percent values are based on Vp(initial).  Or better yet, just report the ratio change (0.20)

Response 19: We apologize for these confusions. As the suggestion we express the Vp/Vp(initial) ratio in decimal.

MODIFIED   p.1 lines 27-33,

p.11 lines 276-277,

p.11 lines 297-302,

p.13 lines 330-331,

p.14 lines 373-379.

Point 20: 322 Figure 16??

Response 20:

MODIFIED p.13 lines 343-344.

Figure 16 shows the wave velocities and displacement plotted over time.”

Point 21: Line 333.  Is it Figure 17 or 15?

Response 21:

MODIFIED p.14 line 356.

“The same wave velocities and displacement data in Figure 16 were plotted in Figure 17.”

Point 22: Figure 3 and the manuscript text create confusion.   In 3a, it appears that each layer is separated in a tray (frame) with wheels (stated in the text).  I assume that the soil column is inside these stacked frames (like a typical shear stack model) – but this needs to be clearer to the reader because the way 3b is drawn makes it confusing.  This is where a photo might help – and better clarity in the manuscript text.  For any reader not familiar with a ring or shear stack, a mis-impression of the apparatus here can make the entire manuscript confusing. 

Response 22: We add two photos as the following.

MODIFIED p.4 lines 118-123.

 (a) A photo overview of the multi-layer shear model; (b) an independent frame of the multi-layer shear model;

Point 23: The authors should also insure that everything mentioned in the abstract is also mentioned in the paper.

Response 23:

MODIFIED.

Round 2

Reviewer 1 Report

The paper can be accepted in the present version after a minor correction.

Minor correction

The correct reference, number [6], is:

Conte, E., Troncone, A., Vena, M. (2017). A method for the design of embedded cantilever retaining walls under static and seismic loading. Géotechnique, Vol. 67, 12, 1081-1089. 

and not  

6. Conte, E.; Troncone, A.; Vena, M. A method for the design of embedded cantilever retaining walls under static and seismic loading. Géotechnique 2017, 1–9.

i.e. page numbers are incorrect.

Reviewer 2 Report

General comments:  I reviewed this manuscript previously and found it needed major revisions.  This very is dramatically improved.  The authors have addressed most of my comments - and the few they did not were suggestions anyway.  The language is dramatically improved as well.

There are still a few minor typos here and there and I think the authors should go through it one last time to look for typos and formatting errors.  Because the revised version I received I was a static (PDF) version with track changes, some of the formatting is changed by the track changes (which I can't turn off).  I think in general, these issue will be very minor.  I have a few specific comments below on obvious things I caught.

Title: "in" is fine. I think "Compress" should be "Compression"?

First sentence of the introduction: Because the sentence is not about a single landslide, "Landslides" (with an s), "occur" (with no s), "cause" (no s)

In section 4.2. Calibration of Soil Moisture Sensors:  there may be some formatting errors regarding paragraphs and indenting.  However, I have a static (PDF) version with track changes - and can't tell for sure.

"7 RESULTS AND DISCUSSIONS"  All caps is not consistent with rest of manuscript.
